# Identifying the Sweet Spot of Padel Rackets with a Robot

**DOI:** 10.3390/s23249908

**Published:** 2023-12-18

**Authors:** Carlos Blanes, Antonio Correcher, Jaime Martínez-Turégano, Carlos Ricolfe-Viala

**Affiliations:** Instituto de Automática e Informática Industrial, Universitat Politècnica de València, Edificio 8G, Acceso D, 3a Planta, Camino de Vera s/n, 46022 Valencia, Spain; carblac1@ai2.upv.es (C.B.); ancorsal@upv.es (A.C.); jaumartu@upv.es (J.M.-T.)

**Keywords:** padel, tactile sensor, system identification, robot sensor, accelerometer, signal process

## Abstract

Although the vibration of rackets and the location of the sweet spot for players when hitting the ball is crucial, manufacturers do not specify this behavior precisely. This article analyses padel rackets, provides a solution to determine the sweet spot position (SSP), quantifies its behavior, and determines the level of vibration transmitted along the racket handle. The proposed methods serve to locate the SSP without quantifying it. This article demonstrates the development of equipment capable of analyzing the vibration behavior of padel rackets. To do so, it employs a robot that moves along the surface of the padel racket, striking it along its central line. Accelerometers are placed on a movable cradle where rackets are positioned and adjusted. A method for analyzing accelerometer signals to quantify vibration severity is proposed. The SSP and vibration behavior along the central line are determined and quantified. As a result of the study, 225 padel rackets are analyzed and compared. SSP is independent of the padel racket shape, balance, weight, moment of inertia, and padel racket shape (tear, diamond, or round) and is not located at the same position as the center of percussion.

## 1. Introduction

Padel is a racket sport with an increasing number of players worldwide and is receiving increasing research interest [1,2]. The International Padel Federation [3] is the principal association formed by 51 national federations representing over 90 countries. Current estimations show over 18 million players and 300 thousand federated players. Spain is a major country in terms of players, with over 4 million players [4]. A professional circuit created by a private firm (World Padel Tour) organizes tournaments in cities around the world (although most tournaments are in Spain). Padel is based on a well-known sport (tennis), but the court is smaller (20 × 10 m) and enclosed. Therefore, the ball can bounce on the boundaries to allow longer plays. These characteristics allow players of all ages and genders to have fun playing padel regardless of their physical fitness [5,6].

The basic element for playing padel is the racket. Because of the growth of padel sport, padel racket sales have increased notably in the last few years [7]. That commercial opportunity has led to the creation of many firms dedicated to padel racket manufacturing. Nowadays, padel rackets are handcrafted, and there is no standard for this manufacturing process. Thus, when a company develops a new racket model, it defines its characteristics based on its experience and commercial interests. This lack of standardization is a problem for users (who have no way of comparing rackets with each other except by direct testing) and for companies because it is complex to determine the characteristics of a particular racket design beyond evaluation by expert users. This problem is currently being focused on by researchers in disciplines ranging from materials technology [8,9,10], to vibration and players’ performance [11], to artificial intelligence. One of the main characteristics that interest both companies and researchers is the sweet spot position (SSP) and its behavior.

Injuries are common in amateur padel players, especially in upper body parts such as the shoulder, the elbow, or the wrist [12,13,14,15]. The stroke technique is essential to avoid those injuries because hitting the ball outside the SSP brings extra effort to the arm. Therefore, in the case of amateur players, information about the wideness of the sweet spot is essential in choosing a racket. Low-level players should choose wide sweet spot rackets to avoid injuries from a bad stroking technique. Traditionally, companies grow their designs to widen the sweet spot region so players can improve their stroking technique.

There is research focused on determining the SSP in different sports rackets. The first documented experiment was carried out by Brody [16,17]. This research addressed the performance of the tennis racket by locating the SSP over the geometry of the racket. Later, Cross [18] presented an experimental method to measure the SSP. He attached some piezo-electric sensors along a tennis racket to measure the propagation of the pulses through the racket. He experimented with the racket suspended and hand-held and threw low-speed balls against different points of the racket. First, he measured the delay from the ball’s impact to the hand and the velocity and force induced in the handle. He concluded that the center of percussion (very similar to SSP) can be uniquely defined in the hand-held case as the rotation axes of the racket are restricted. There has yet to be a consensus about the racket SSP definition. Cross [19] summarized that, in theory, the SSP is affected by vibrational feeling, ball energy, center of percussion, and vibrational modes.

Ref. [20] presented a work addressing the experimental SSP point estimation in beach tennis rackets. This approach focuses on modal analysis to study the frequency response of the racket. He suspended the racket with soft bungee cords to approximate a non-restricted movement. Then, he hit the racket with an impact hammer and measured the frequency response functions on 18 measurement points. One interesting conclusion is related to the boundary conditions regarding the tests. He restricted the racket’s movement with a clamp and compared the results with an unrestricted response. He found a high correlation in the results, although he claimed that the clamped test needed time to tune the stiffness in the clamp. The study does not consider the influence of hand-held.

Ref. [21] studies the SSP as the center of percussion of a simple wood beam with two accelerometers. This research identifies the influence of the SSP location produced for the hand-held and notes how modern rackets are “so stiff and light that all vibration modes are suppressed, even the fundamental one”. Sarkar [22] uses accelerometers at the players’ wrists to determine the SSP of cricket bats. The SSP is located where the sensor’s response produces the lowest value.

Ref. [23] reviews different studies to determine tennis racket features. It defines the SSP as a mixture of three factors: “the area of maximum rebound ball speed, the point of minimum vibration (node), and the point of no frame reaction (center of percussion, COP)”.

As padel is a novel sport (although it was born in the 70s, its practice did not become popular until 20 years ago), there is still not much research focused on the physical properties of padel rackets [24], and no one is studying the SSP. This paper proposes a methodology to measure the padel rackets’ SSP and amplitude. The paper also studies padel racket vibration behavior, or tactile response, when padel rackets are hit along their central line. The methodology requires specific clamps to fit padel rackets, a calibration fitting process, and a robot that hits the padel racket along its central line with high precision (less than 0.1 mm), ensuring repeatability and reproducibility. The experimental test has been done for 225 padel rackets, and the results are correlated with balance, weight, and moment of inertia.

## 2. Measurement Process

Padel rackets are hit at different points along their central line, and the severity of every impact is quantified with vibration measures. An impact on the SSP generates the minimum reaction over the grip of a padel racket. This research explores this effect by hitting the racket and observing the effect on the grip. The process generates impacts along the rackets’ central line with a robotic aid. A cradle ensures the same initial racket position for every hit, independently of where the hit is produced. Several sensors collect the severity of every hit.

The cradle (Figure 1) allows hitting every padel racket and keeping it in the same position after each impact. The cradle rotates around an axis X located in the central plane of every racket and at the racket bottom. This way of cradle-holding the racket approximates a player’s hand-hold. Two shock absorbers hold the cradle with every padel racket in place, keeping it in the same position after each impact. A pneumatic guided cylinder (SMC MGPL12_125Z, SMC, Chiyoda-ku, Tokyo, Japan) ensures it remains in the original position after each impact by pressing it against the two end stops. The padel racket handle is fixed to the cradle with a clamp attached with four screws DIN 912 M5x30 at 2 Nm. Before fixing the padel racket, two end stops allow for leveling it horizontally. The clamp has accelerometers attached.

### 2.1. System Calibration

A specific regulation determines the maximum dimensions of padel rackets [25]. Despite this regulation, the dimensional variability of the padel rackets is significant, making it difficult to adjust the fixing system. Therefore, it is necessary to carry out a calibration process that guarantees the correct placement of the padel rackets. Each new padel racket must have the same relative cradle–robot position when the assembly is done. The first step is to attach the padel racket to the cradle with the clamp. A specific tool makes it possible to guarantee the same padel racket and sensors’ relative position in every new assembly. After that, the racket should be horizontally leveled using two stops at the padel racket’s lateral ends. Now, the racket’s flat surface is parallel to the floor. With this process, the plane XY of the racket is parallel to the robot tool plane XY but not yet the vertical Z rotation. The robot tool reflects the padel racket surface and moves within a horizontal plane parallel to the padel racket surface. Both the Z-axis, the robot, and the racket are parallel.

The second step is to reference the padel racket position concerning the robot’s reference system. In this process, the padel racket’s vertical rotation (z-axis) concerning the robot is defined by determining the padel racket’s central line. A calibration tool is fitted to the robot, and with two points, the operator manually defines the racket central line. This process ensures that the XYZ axis of the racket and robot tool is parallel. When the central line is defined, it is necessary to reference the starting point. This point is in the racket’s center line, at 40 mm from its top and vertically at 35 mm. The calibration tool has a plane located at 40 mm from the robot tool center point (TCP), and the operator moves the robot to this plane locating (x position), aligns the center line (y position), and moves 35 mm up the TCP (z position) to define the starting point.

This methodology ensures that the pneumatic piston’s speed consistently hits the padel racket at the same velocity.

### 2.2. Robot Operation and Process

The Stäubli TX60 (Pfäffikon, Switzerland) robot ensures high process repeatability (0.02 mm) to hit the padel racket at different points. The calibration process and the calibration tool guarantee the robot’s relative position, defining the normal plane of the padel racket, the longitudinal axis, and the starting point 40 mm from its top. The robot moves the tool composed of a low-friction pneumatic impacting cylinder (Figure 1, SMC MQQTB16_40D, Chiyoda-ku, Tokyo, Japan), feed at 4 bar pressure, and the round plastic part that impacts against the padel racket 90 times along its central longitudinal axis. This process is repeated three times. The first impact is located at 40 mm from the padel racket top, and the last is located at 220 mm.

In every impact (Figure 2), the robot moves 2 mm along the padel racket longitudinal axis, deactivates the pressing cylinder, puts out a low digital signal connected to the acquisition card (NI USB 6210, National Instruments, Austin, TX, USA), activates the impacting cylinder, waits 0.5 s, deactivates the impacting cylinder, change to high the signal, activates the pressing cylinder and repeat. Figure 2 illustrates the overall process of the robot program and the devices to control it.

One gyroscope (ADXRS642, Analog Devices, Wilmington, MA, USA) and one analog accelerometer (ADXL278, Analog Devices, Wilmington, MA, USA), +/−500 m/s^2^ range, are attached to the clamp that grasps the padel racket to the cradle. Both are not in direct contact with the handle of the padel racket. The gyroscope collects the angular rotation speed (Rx), and the accelerometer collects the vertical acceleration (*Az*(*t*)). Angular speed rotation and acceleration are sent to a computer using a NI USB-6210 A/D data acquisition module. Signals are sampled at 30 KHz and low-pass filtered at 1500 Hz. A software programmed in LabVIEW™ (LabVIEW 2022 Q3, National Instruments, Austin, TX, USA) for signal acquisition collects the accelerometer data from the fall-down trigger 8000 samples. Only the signal from the contact starts till 1200 samples are collected. Figure 3 is an example of acceleration registered when one padel racket is hit at different positions along its longitudinal center line. Acceleration severity decreases approximately in the middle of this line. A similar perception has a player when a ball impacts against the padel racket while holding it. Shaking perception is the minimum when the ball hits the SSP.

Acceleration signals are processed following the developed procedure [26]. In this procedure, accelerometer signals are used to recognize the firmness of fruits and vegetables while they are hit with robot gripper fingers. In this case, the accelerometer is located at the bottom of the padel racket handle. The procedure processes accelerometer (*Az*(*t*)) signals to provide the parameter *V*:(1)$$V=\frac{Energy}{mass}=\int_{t0}^{t1}{Az(t)}^{2}\times dt$$

### 2.3. Design Validation and Test

*V* parameter is evaluated in every impact from the start to the endpoint (Figure 3), where *t*_0_ is the starting contact point, and *t*_1_ is 0.04 s. Ninety values of *V* are got across the central line of padel rackets. This process is repeated three times to get the average values.

After each impact on the face of the padel racket, the data acquisition system collects the accelerations from the accelerometer located at the end of the padel racket handle, where the player’s wrist is. The signal measures the severity of the impact received on the player’s wrist. The same experiment with a diameter of 34.5 mm and 545 mm-long beechwood allows us to define its SSP as the minimum value of *V* (Equation (1)). To locate the SSP, *V* is evaluated at various points along the bar to assess the severity of the impact when the bar pivots around an axis located at the end. Figure 4 illustrates the evolution of parameter *V* from 40 to 220 mm measured from the end of the bar. The figure displays three types of values:The smoothed value of *V* by a cubic spline curve;The unfiltered value of *V*;The amplitude of the first sinusoidal signal after the Fast Fourier Transform (FFT).

The unfiltered and amplitude signals oscillate periodically near the smoothed one, showing changes in slope. In this configuration, the SSP of the bar is the minimum value of *V* located at 124 mm from the end of the bar. Acceleration severity, parameter *V*, increases as we move away from the SSP.

The shape of the acceleration signals (Figure 5) between close points exhibits repetitive patterns. Typically, the values of acceleration peaks/valleys change between close points while the shapes remain similar. Figure 5 shows the accelerometer signal when an impact occurs at points located at 84, 98, and 124 mm. The SSP is located at 124 mm. The distance from the SSP and the point at 84 mm is higher than at 98, in the same way the *V* value is higher at 84. The severity of the impact accelerations at 84 mm is significantly higher; the absolute peak values are greater than those of the signal at 98 mm. Several phases are identified in Figure 5:Phase 1 contains the direct impact (shock). 84 and 98 have some regular shapes in this phase, which differs from the SSP with more irregular shapes;Phase 2 depends on the free motion following the bar’s harmonic. In this phase for 84 and 98, accelerometer signals have similar patterns, only changing the values of the peaks/valleys. The accelerometer signal at SSP is minimal, and no pattern can be recognized.

During phase 2 at 84 and 98, two harmonics are observed, corresponding to the frequencies of the first two oscillation modes of the bar. The higher frequency (f2) is consistently evident at 84 and 98 mm but not at the SSP. The lower frequency (f1), which is lower in amplitude, oscillates significantly depending on the impact point.

Analyzing phase 2 in detail, the peak values of hit at 98 are higher than those corresponding to 84. Impacts in the middle of the node’s distance for f1 harmonic motion will excite this mode more, while impacts in the nodes of this harmonic node will not excite this mode. The impact at 84 is closer to the nodes of the f1 harmonic motion than the impact at 98.

The values of *V* (Figure 4) at 84 and 98 mm are above the smoothed values of *V*, and this difference is more pronounced at 98. For those non-smoothed *V* values significantly lower than the filtered signal, the second phase of the impact exhibits more attenuated signals, and the oscillation at frequency f1 is not observed. These points are closer to the nodes of the bar’s motion at frequency f1.

Figure 6 shows the amplitude and phase for the first sinusoidal after the FFT of each acceleration signal at each impact point. This graph illustrates a sign change in phase direction between 132 and 134 mm due to a change in the beginning of the motion sense. The accelerometer detects the vertical motion changes even when the beechwood bar pivots around a point that theoretically cannot move vertically. This change in sense represents the location of the center of percussion [21], which does not coincide with the SSP located at 124 mm. The phase values of the first sinusoidal change depend on the impact position and are close to zero between 136 and 150 mm. In this area, the start sense of the bar is not clearly defined, and accelerometer signals have low values. Between the region from 40 to 130 mm, the bar starts down. Between 132 and 150 mm, there is a change in sign, and from 152 mm onwards, the bar starts up. The center of percussion will be located between 132 and 150 mm.

### 2.4. Padel Racket Measurements and Analysis

The SSP is located where the *V* value is the minimum (V_min_). The SSP is measured from the top of the padel rackets because the manufacturers use different plastic taps with different thicknesses, even for the same padel racket.

In Figure 7, for the 225 padel rackets analyzed, all SSPs are located between 118 and 147 mm with an average position of 134 mm. There is a tendency to increase the V_min_ value when the SSP is higher. This tendency is clear for average values with a high coefficient of determination (R^2^ = 0.88) but not when all cases are analyzed together (R^2^ = 0.15).

Figure 8 shows the variation of SSP with the balance. The balance is the position of the center of gravity and has been measured according to the procedure described in [14]. In this case, it is not possible to correlate the balance with the SSP. This result is not according to [21], where the center of percussion of a bar increases with the square of its length.

Similar studies have been done for the weight, moment of inertia around the axis X, and shape (Figure 9 and Figure 10). No correlation was found between SSP and padel racquet weight (Figure 9), even for averaged values. Similar conclusions were found between SSP and moment of inertia Ix (Figure 10). The moments of inertia Ix of the padel racquets were measured according to [24].

Figure 11 presents a study involving 93 different padel racket shapes, categorized into three groups: diamond (3), tear (2), and round (1). The research aims to explore the correlation between these shapes and the SSP. In each small dot point, the first number denotes the specific racket shape, while the second is the average count of padel rackets associated with that particular shape. Additionally, each large dot point indicates the average shape for each SSP category.

Despite the importance of padel racket shape for players [13], no significant correlation was found between racket shape and the SSP, even for the shape average values.

Figure 12 illustrates the average *V* value for 225 padel rackets along its central line and its standard deviation when impacted along its central line. Values around the SSP have less standard deviation than the extreme values, and this effect increases for positions close to the handle. Figure 12 shows that the average SSP value is 134 mm from the top of the padel rackets. The variation in the behavior is significant and could be due to the different shapes and materials.

Figure 13 shows how different the behavior of padel rackets could be when analyzing the *V* parameter. Figure 13 shows the *V* curve of six different padel rackets. The six padel rackets show a minimum value of around 134 mm between 128 and 142 mm. P6 or P3 is the classical shape of a padel racket for beginners. It has low *V* values, which involve low vibration transmitted to the wrist despite the impact point in the racket. Moreover, the SSP zone is wide. P5 seems to have a similar behavior, but it peaks around 90 mm. When the player hits the ball at that position, he will feel a different hit behavior. P2 has a wide sweet spot, but any hit outside the sweet spot zone will fall very differently from a hit on the racket’s SSP. P1 and P4 show classical behavior for advanced padel rackets. The sweet spot zone is narrow, so the player must hit the ball precisely. Moreover, the *V* value is generally higher, so the player needs to be physically fit to play with these rackets.

Regarding the shapes, only the P2 signal has 1 clear valley, while the others have 2 or 3 valleys, the first close to the top, approximately at 60 mm, and the last close to the handle, between 170 and 210. P2 has a U SSP area shape, while P4 is an example of a *V* SSP area shape. The minimum *V* value (V_min_, *V* value at the SSP) is similar for P2 and P3, but their behavior differs significantly from the SSP area. Generally, padel rackets with low V_min_ have lower *V* values along the racket, but this is not a rule out of the SSP. Some padel rackets have a wide SSP area, as in P3, where *V* values around V_min_ are similar from 94 to 180 mm. Some padel rackets have narrow SSP behavior, as in P4.

## 3. Discussion

There has yet to be a consensus on the definition of the SSP. This article proposes the parameter *V* to analyze each impact and define the SSP where *V* is the minimum. *V* characterizes the severity of impacts where the player’s wrist is. It is the shaking perceived by the player. *V* depends on the impact point’s location, impact shock, and the vibration of the free modes of the padel rackets. 

With a round beechwood bar, acceleration signals have two phases: the shock of the impact dominates the first, and the main bar’s free vibration modes dominate the second. The second phase of the impact is influenced by the location of the nodes of the bar’s harmonic movements. Acceleration signals follow a similar pattern between close impacts, with minor peaks/valleys value changes. Between close impacts, if the peaks/valleys of the first phase increase, that does not mean that the peaks/valleys of the second phase increase. For this reason, parameter *V* exhibits oscillations around its mean values. When an impact occurs between harmonics nodes, the excitation of the bar in this mode is maximal; if the impact is at the nodes, the excitation in this mode is null. A detailed study of the FFT accelerometer signals in phase makes it possible to identify the position of the center of percussion, and this position is not located at the SSP, where *V* is the minimum.

A padel racket is a complex object with various shapes, frame configurations, handle types, and materials (carbon fibers, fiberglass, and ethyl vinyl acetate rubber). Identifying the different phases of impact, free motion harmonic frequencies, amplitudes, and phases is challenging and unclear. However, it is possible to assess the severity of the impact through the parameter *V*. 

From the point of view of padel players and manufacturers, the shape is one of the most characteristic features of the padel rackets. Traditionally, it has been associated with the playing style the players want to develop. The most common shapes are diamond, teardrop, and round. The shape directly affects the mass distribution, so diamond rackets are expected to have a higher balance, teardrop rackets have a medium balance, and round rackets have a lower balance. Power shots are favored by a high balance in rackets, and control shots by a low balance in rackets, so attack players prefer diamond rackets and defensive players prefer round rackets. Teardrop rackets remain in a polyvalent range.

Nevertheless, balance (an inertia) is also affected by factors other than the shape. The different materials used on the different parts of the racket, the heart shape, the heart mass, and others, are factors that highly modify the balance and inertial properties of rackets, so it is not always true that the shape of the racket makes it suitable for a playing style. Therefore, the shape is less important in selecting a racket than the physical features of each playing style.

The results of this research could help develop new technologies in padel rackets. The test we propose in this paper is useful for testing the effect of new racket shapes, materials, and manufacturing techniques (such as 3D printing) in the SSP. Manufacturers can take profit from the SSP measuring device to study any improvement in the racket. Moreover, some manufacturers provide custom weighting in the racket by attaching weights to the frame. The scientific effect of the SSP could be tested with the methodology we propose.

One of the conclusions of this paper is that more research has to be done. It is necessary to study each padel racket to determine the SSP and its behavior upon impact. The high variability in materials, manufacturing processes, or shapes makes it complicated to develop offline models of the SSP. Manufacturers will benefit from more knowledge concerning the effects of changes in their designs of the SSP before manufacturing the rackets. To advance these models, more collaboration between researchers and manufacturers will be needed. The sharing of information between the stakeholders (scientists, players, and manufacturers) will improve the analysis of the data results in the experiments. Moreover, including measures in the manufacturing line will help advance quality control in the factories.

It is also important to address the gap between marketing and scientific results. If the racket manufacturers start scientifically measuring their racket characteristics (such as the SSP), the player community will benefit from including indicators (such as the *V* parameter) in the sport. Manufacturers could use these as marketing highlights that include scientific measures of the racket characteristic, gaining credibility. The design of courses for players and couches to explain the physical characteristics of the racket could be an added value to the product.

## 4. Conclusions

The severity of the impacts in different padel rackets has been evaluated and studied with the *V* parameter. *V* uses the information of an accelerometer fitted in a cradle at the bottom of the padel rackets.

This study shows no clear correlation between SSP and the other classical characteristics of the racket (mass, balance, inertia, and shape). Therefore, the commercial identification of round rackets with wide sweet spots has proven uncertain. The complexity of the rackets, integrating different materials, hole patterns, and bridge elements, among other elements, and the manual manufacturing process make it challenging to foresee the SSP and wideness in a design phase. So, the only way of determining the SSP is by measuring it. This paper has shown a method and a device to measure the SSP, the SSP wideness, and quantify the severity of each hit on the wrist of the player. The method also allows the comparison of the behavior of padel rackets with the help of the *V* parameter.

Applying the same methodology and conducting a detailed and in-depth analysis of a round beechwood bar, it is possible to determine that the position of the center of percussion does not coincide with the position of the sweet spot.

## Figures and Tables

**Figure 1 sensors-23-09908-f001:**
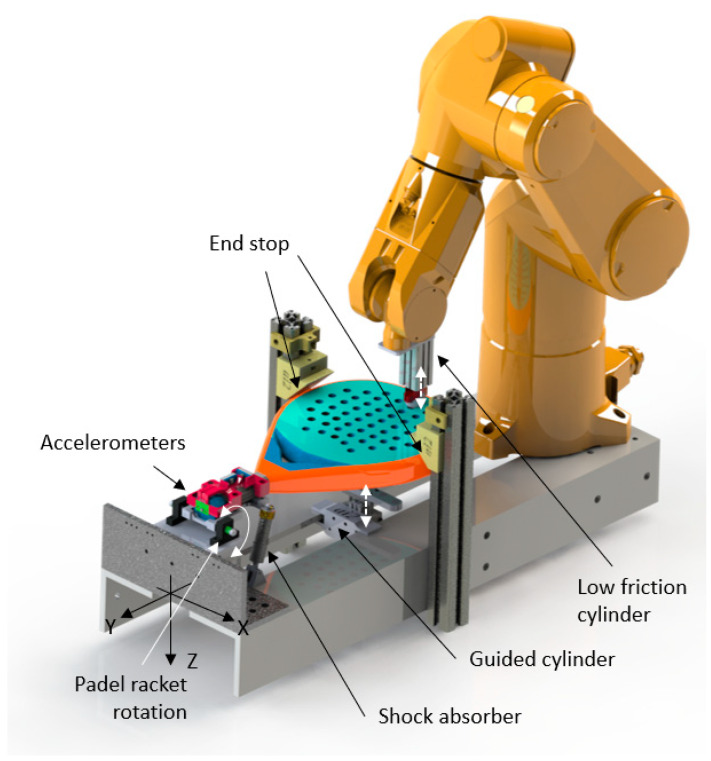
Robot, cradle motion, and devices.

**Figure 2 sensors-23-09908-f002:**
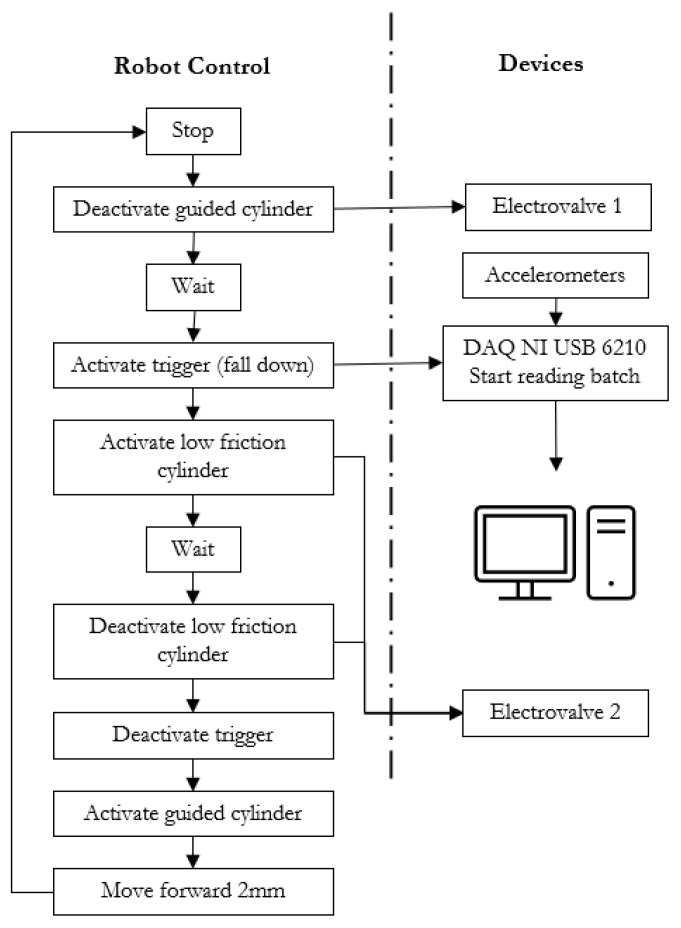
Robot process flow and devices with their communications for every hit.

**Figure 3 sensors-23-09908-f003:**
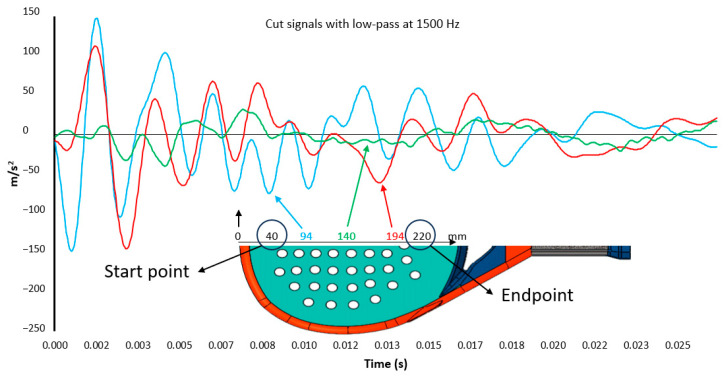
Example of accelerometer response when padel racket is hit at different points.

**Figure 4 sensors-23-09908-f004:**
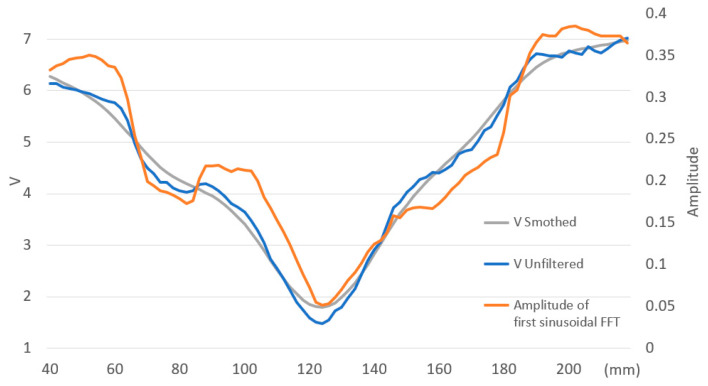
Evolution *V* unfiltered and smoothed, and amplitude of the first sinusoidal after FFT for a round beechwood bar.

**Figure 5 sensors-23-09908-f005:**
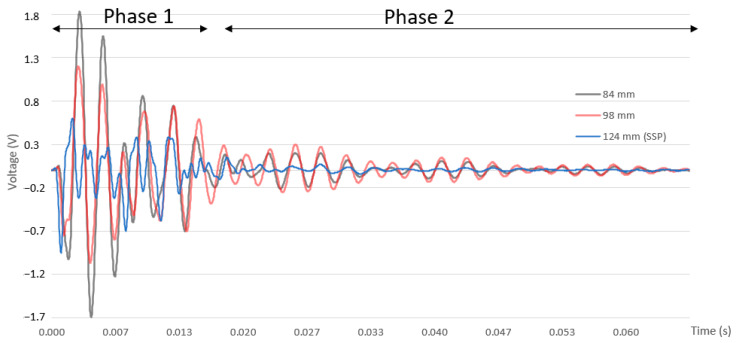
Accelerometer response in a round beechwood bar at 84, 98, and 124 mm.

**Figure 6 sensors-23-09908-f006:**
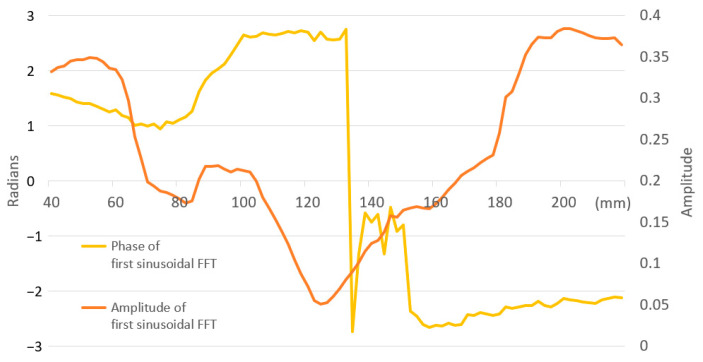
Amplitude and phase for the first sinusoidal every impact in a round beechwood bar wood after FFT.

**Figure 7 sensors-23-09908-f007:**
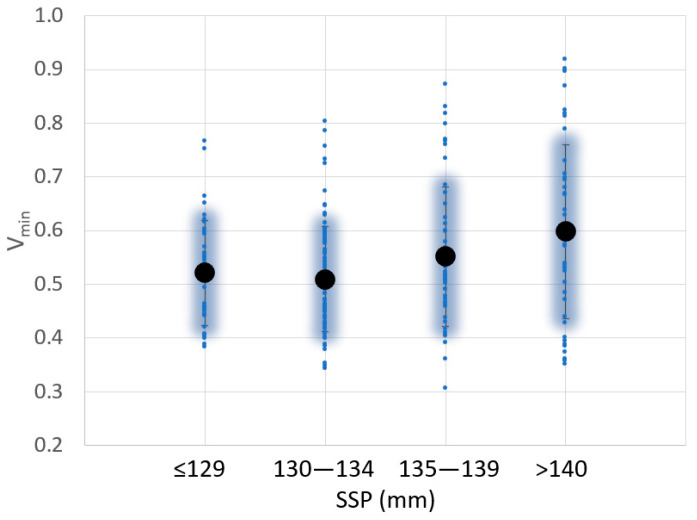
Variation of the sweet spot position (SSP) and V_min_ value for 225 padel rackets analyzed.

**Figure 8 sensors-23-09908-f008:**
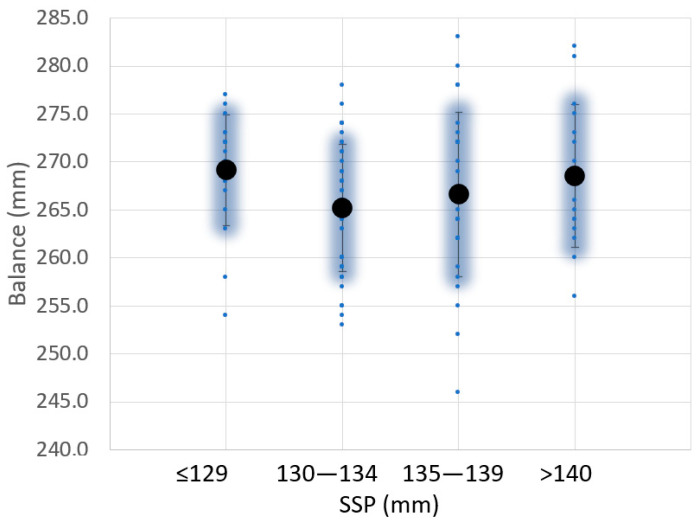
Variation of the sweet spot position (SSP) and balance value for 104 padel rackets analyzed.

**Figure 9 sensors-23-09908-f009:**
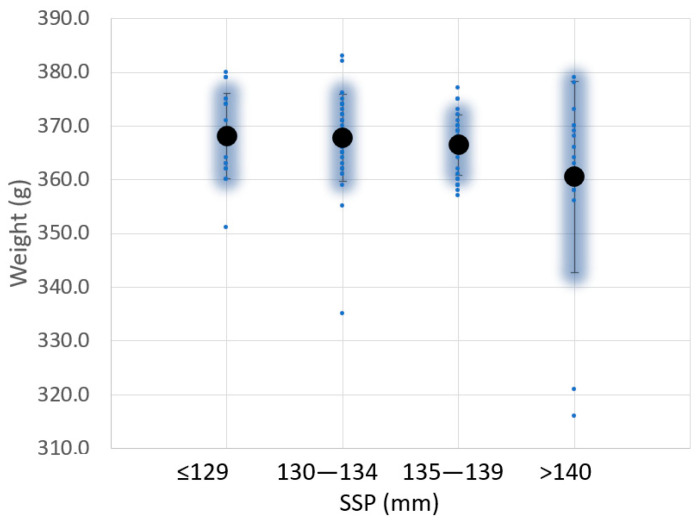
Variation of the sweet spot position (SSP) and weight value for 104 padel rackets analyzed.

**Figure 10 sensors-23-09908-f010:**
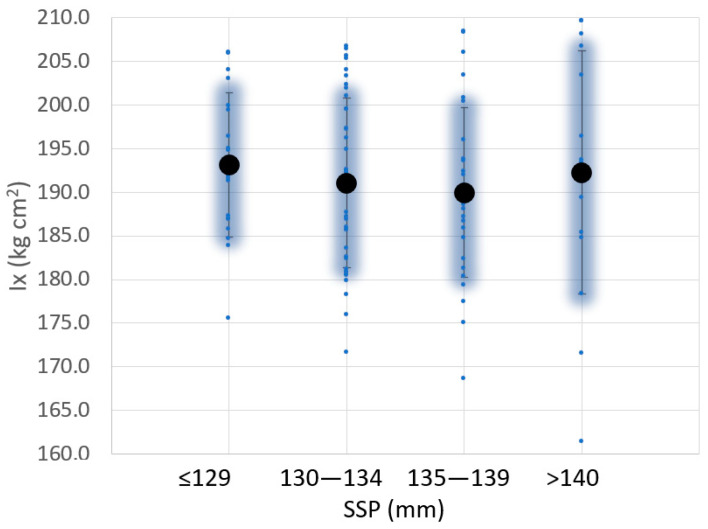
Variation of the sweet spot position (SSP) and moment of inertia around the axis X for 104 padel rackets analyzed.

**Figure 11 sensors-23-09908-f011:**
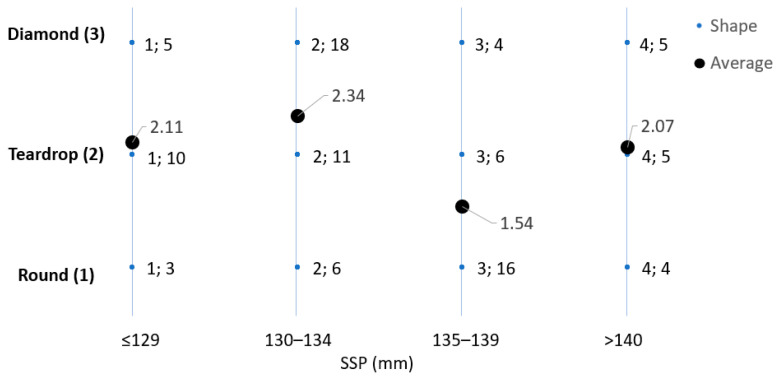
Variation of the sweet spot position (SSP) and padel racket shape for 93 padel rackets analyzed.

**Figure 12 sensors-23-09908-f012:**
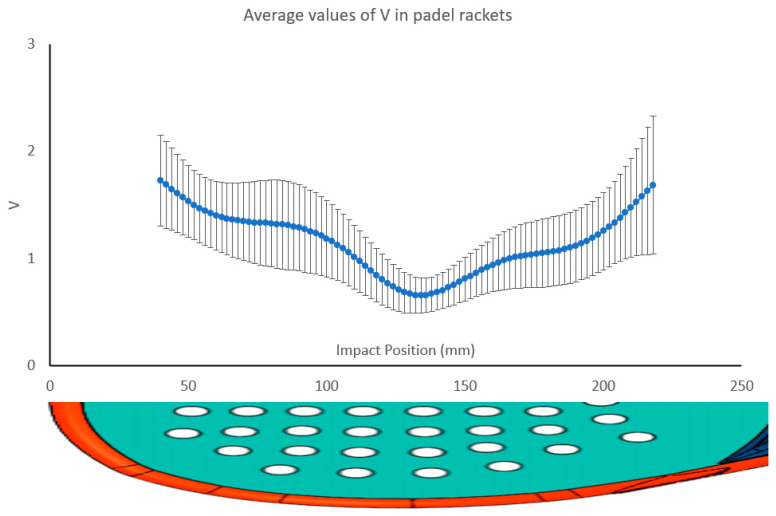
Variation of the *V* value for 225 padel rackets along its central line and standard deviation in every impact point.

**Figure 13 sensors-23-09908-f013:**
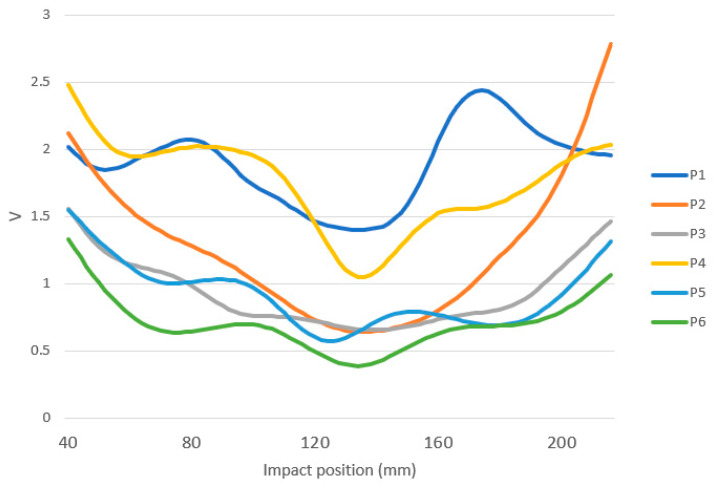
Example of *V* values for six different padel rackets.

## Data Availability

Restrictions apply to the availability of these data, which were used under license from Testea Padel Laboratory for this study.

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
