# Peer review of "Identifying the Sweet Spot of Padel Rackets with a Robot"

_sensors, 2023, doi:10.3390/s23249908_

Round 1
Reviewer 1 Report
Comments and Suggestions for Authors
1. Authors need to improve the abstract and should include some values of the results as well.
2. Authors should also separate the discussion and conclusion section. 
3. In the keyword, authors also included tactile sensors. Is it also used in experiment? If so, explain that. 
4. Authors should also improve the references section. There are some suggestions provided below: 
a. A. Fathema et al., IEEE Sensors Journal 21, no. 7 (2021): 9546-9552.
b. M. M. Hussain,et al. (2022) "Angular accelerometer device and method based on capacitive sensing." U.S. Patent No. 11,243,227.
c. D. V. Thiel et al., .Swing profiles in sport: An accelerometer analysis. Procedia Engineering, 72, 624-629, 2014.
d. M. P. Smith et al., Scientific reports 8, no. 1 (2018): 15055.
e. M. A. I. Anik et al., 2016 19th International Conference on Computer and Information Technology (ICCIT), Dhaka, Bangladesh, 2016, pp. 213-217.
f. G. Aroganam et al., Sensors, 19(9), p. 1983, 2019.
g. J. F. Wagner, Gyroscopy and Navigation, 9, 1-18, 2018.
h. R. B. Mishra et al., Advanced materials technologies, 6(4), 2001023, 2021.
5. How many padel rackets authors has experimentally characterised, give the name and manufacturing company as well.
Reviewer 2 Report
Comments and Suggestions for Authors
The authors of the study delve into the analysis of padel rackets, underscoring the significance of factors like vibration, sweet spot location, and their impact on gameplay and injury prevention. They present a method to determine the sweet spot's position and quantify its behavior, introducing a parameter (V) for assessing the severity of impacts on the player's wrist. Utilizing accelerometers and a robot, the authors analyze vibration behavior, comparing over 150 rackets in the study. The complexity of padel racket design, including shape variations and material choices, challenges traditional associations between racket shape and playing style. The authors argue that there is no clear correlation between sweet spot position and other racket characteristics, advocating for measurement over design assumptions. They emphasize the need for further research on racket materials and their impact behavior along the central line to establish correlations with the proposed V parameter.
The manuscript is very well written and the experimental methodology seems resealable. As authors explain, SSP is hard to define but for future studies, it'd be great if the authors can correlate the SSPs found in this study to actual performance of racket. 
Comments on the Quality of English Language
The manuscript is very well written and easy to read. There are minor editorial changes that need to be corrected before publication.
Reviewer 3 Report
Comments and Suggestions for Authors
Nice work.
Some suggestions:
1. Experiment with Different Racket Shapes:
Since the study highlights the impact of racket shape on balance and playing style, manufacturers and players could experiment with different shapes to better understand their impact on the game. Exploring unconventional shapes might lead to the discovery of new characteristics that enhance performance.
2. Material Innovation:
The study mentions the influence of materials on racket properties. Manufacturers could invest in research and development to explore new materials or combinations that can optimize the balance, mass distribution, and overall performance of padel rackets. This could involve testing materials not traditionally used in racket construction.
3. Advanced Manufacturing Techniques:
Given the complexity of racket design, exploring advanced manufacturing techniques, such as 3D printing or precision engineering, might help improve consistency in producing rackets with desired properties. This could contribute to a more predictable outcome in terms of SSP, wideness, and other characteristics.
4. Player-Specific Racket Customization:
Recognizing that playing styles vary, manufacturers could offer more customized racket options tailored to specific player preferences. This could involve considering individual factors such as playing style, wrist strength, and impact behavior, allowing players to choose rackets that suit their unique needs.
5. Data Sharing and Collaboration:
Encourage collaboration between manufacturers and researchers to share data on padel racket materials and their impact behavior. This collaborative effort could lead to a more comprehensive understanding of the correlations between materials, minimum V values, and impact behavior along the central line of the racket.
6. Education and Marketing:
Since the study challenges the traditional association of round rackets with wide sweet spots, there is an opportunity for manufacturers to educate consumers about the complexities involved. Marketing strategies can focus on promoting the importance of measuring SSP, wideness, and impact severity, providing players with a more informed approach to selecting rackets.
7. Continuous Measurement and Improvement:
Implementing a continuous measurement process during and after manufacturing could ensure that each racket meets the desired specifications. This could involve integrating the developed method and device for measuring SSP and impact severity into the manufacturing process to ensure consistency and quality control.
8. Further Research on Padel Racket Materials:
Given the suggestion that further studies considering padel racket materials would help identify correlations, invest in research to fill the data gap. Understanding how specific materials influence the minimum value of V and impact behavior can lead to more informed material choices in racket construction.
9. Player Feedback Integration:
Encourage players to provide feedback on racket performance, especially concerning SSP and impact behavior. This user-generated data can be valuable for refining racket designs and identifying areas for improvement that may not be apparent in controlled laboratory settings.
10. Development of Training Programs:
Develop training programs for players and coaches to understand and utilize the information provided by the V parameter. This could help players optimize their playing styles based on racket characteristics, leading to improved performance on the court.
Round 2
Reviewer 1 Report
Comments and Suggestions for Authors
Accepted